# Automatic Segmentation of Retinal Fluid and Photoreceptor Layer from Optical Coherence Tomography Images of Diabetic Macular Edema Patients Using Deep Learning and Associations with Visual Acuity

**DOI:** 10.3390/biomedicines10061269

**Published:** 2022-05-29

**Authors:** Huan-Yu Hsu, Yu-Bai Chou, Ying-Chun Jheng, Zih-Kai Kao, Hsin-Yi Huang, Hung-Ruei Chen, De-Kuang Hwang, Shih-Jen Chen, Shih-Hwa Chiou, Yu-Te Wu

**Affiliations:** 1Institute of Biophotonics, National Yang Ming Chiao Tung University, 155, Sec-2, Li Nong Street, Taipei 112304, Taiwan; max870121@gm.ym.edu.tw (H.-Y.H.); zkkao@gm.ym.edu.tw (Z.-K.K.); 2School of Medicine, National Yang Ming Chiao Tung University, Taipei 112304, Taiwan; ybchou@vghtpe.gov.tw (Y.-B.C.); ray007chen@gmail.com (H.-R.C.); m95gbk@gmail.com (D.-K.H.); sjchen96@gmail.com (S.-J.C.); 3Department of Ophthalmology, Taipei Veterans General Hospital, 201, Sec-2, Shipai Rd., Taipei 112201, Taiwan; 4Department of Medical Research, Taipei Veterans General Hospital, Taipei 112201, Taiwan; cycom1220@gmail.com (Y.-C.J.); hyhuang21@vghtpe.gov.tw (H.-Y.H.); 5Big Data Center, Taipei Veterans General Hospital, Taipei 112201, Taiwan; 6Institute of Clinical Medicine, National Yang Ming Chiao Tung University, Taipei 112304, Taiwan; 7Institute of Pharmacology, National Yang Ming Chiao Tung University, Taipei 112304, Taiwan; 8Brain Research Center, National Yang Ming Chiao Tung University, Taipei 112304, Taiwan

**Keywords:** optical coherence tomography segmentation, deep learning, diabetic macular edema, visual acuity

## Abstract

Diabetic macular edema (DME) is a highly common cause of vision loss in patients with diabetes. Optical coherence tomography (OCT) is crucial in classifying DME and tracking the results of DME treatment. The presence of intraretinal cystoid fluid (IRC) and subretinal fluid (SRF) and the disruption of the ellipsoid zone (EZ), which is part of the photoreceptor layer, are three crucial factors affecting the best corrected visual acuity (BCVA). However, the manual segmentation of retinal fluid and the EZ from retinal OCT images is laborious and time-consuming. Current methods focus only on the segmentation of retinal features, lacking a correlation with visual acuity. Therefore, we proposed a modified U-net, a deep learning algorithm, to segment these features from OCT images of patients with DME. We also correlated these features with visual acuity. The IRC, SRF, and EZ of the OCT retinal images were manually labeled and checked by doctors. We trained the modified U-net model on these labeled images. Our model achieved Sørensen–Dice coefficients of 0.80 and 0.89 for IRC and SRF, respectively. The area under the receiver operating characteristic curve (ROC) for EZ disruption was 0.88. Linear regression indicated that EZ disruption was the factor most strongly correlated with BCVA. This finding agrees with that of previous studies on OCT images. Thus, we demonstrate that our segmentation network can be feasibly applied to OCT image segmentation and assist physicians in assessing the severity of the disease.

## 1. Introduction

Diabetic retinopathy (DR) is a common cause of vision loss in patients with diabetes [1]. Some patients with DR eventually develop diabetic macular edema (DME), an accumulation of fluid in the retina, which typically reduces visual acuity (VA). In clinical practice, optical coherence tomography (OCT) has been widely used to detect DME and classify its components [2,3]. At present, commercial OCT devices can automatically segment the retinal layers and measure the central subfield thickness (CST). Although CST is a key factor affecting a patient’s best corrected VA (BCVA), other factors influence BCVA more and are highly related to disease activity, such as the volume of intraretinal cystoid fluid (IRC) and subretinal fluid (SRF) and disruption of the ellipsoid zone (EZ), part of the photoreceptor layer in the retina [4,5]. However, manually measuring these factors by using OCT images is time- and labor-intensive. A solution to this problem may be fully automatic segmentation for detecting and quantifying microstructural changes in retinal OCT images.

Researchers have developed various algorithms for the segmentation of normal retinal layers from OCT images [6]. However, some microstructural changes, such as those of IRC, SRF, and the EZ, are difficult to quantify because of speckle noise, size and shape variation, and the inconsistent curvature of the retina. 

With the increasing power of computers and sophistication of deep learning algorithms, researchers have attempted to develop artificial intelligence (AI) models that can process OCT images, color fundus photographs and optical coherence tomography angiography (OCTA) for classifying diseases, staging diseases, and extracting features [7,8,9,10]. Some research have even used ophthalmic images and deep learning methods to detect other diseases, such as kidney and cardiovascular diseases [11,12]. Various fully convolutional neural networks (FCNs) [13], such as U-net [14] and SegNet [15], have been proposed to segment the retinal fluid from OCT images [16,17,18,19]. Schlegl et al. developed an FCN method that robustly and accurately segments and quantifies IRC and SRF (area under the receiver operating characteristic (ROC) curve (AUC) of 0.94 and 0.91, respectively) [20]. Fauwl et al. developed a U-net-based model that segments many retinal OCT biomarkers and demonstrated its practicability in classification for clinical referrals [21]. Hassan et al. proposed SEADNet and achieved extremely high accuracy in fluid segmentation. They added dense atrous convolution and atrous spatial pyramid pooling in the bottleneck of the model to extract more features [22]. Several studies also used deep learning as a tool to measure fluid volume to assess the outcome of a clinical trial [23,24]. Ma et al. proposed a deep learning method for segmenting the nerve fiber layer thickness from OCT images [25]. Several studies focused on segmenting the EZ layer. Orlando et al. developed a deep learning method for detecting EZ disruption [26]. They used amplified-target loss to keep the model focused on the center because EZ disruption occurs mainly in the center. They published another study that developed a Bayesian deep learning-based model on the same task, achieving satisfactory accuracy for age-related macular degeneration (AMD) but low accuracy for DME and retinal vein occlusion (RVO) [27]. Silva et al. proposed a deep learning-based algorithm to detect EZ disruption to screen for hydroxychloroquine retinal toxicity [28].

Nevertheless, most papers focused mainly on segmentation methods, lacking a correlation with visual acuity. In this study, we developed a deep learning-based automatic segmentation algorithm to segment the IRC, SRF, and EZ. We evaluated the correlations of these segmented factors and CST, which are used in current clinical practice, with BCVA. We demonstrated that the results of our fully automatic segmentation were consistent with the previous manual segmentation research on OCT features that correlated with BCVA [4].

## 2. Materials and Methods

This study proceeded in two stages. First, we constructed our AI segmentation model; second, we evaluated the correlation of each segment with BCVA (Figure 1). All images were divided into the build model (BM) data set (203 images) or evaluated clinical relationship (ECR) (3084 images) data set. The BM data set was used to establish and validate the model, and the ECR data set was used to evaluate the correlations between the AI-quantified biomarkers and BCVA.

### 2.1. Ethical Approval and Data Collection

Patients with DME who visited the Department of Ophthalmology of Taipei Veterans General Hospital (TVGH) for OCT imaging at any time between February 2011 and August 2019 were enrolled. This study was approved by the Institutional Review Board of TVGH in Taipei, Taiwan. OCT images (HD-OCT 4000, Carl Zeiss Meditec, Dublin, CA, USA; RTVue-XR Avanti, Optovue, Fremont, CA, USA), CST measurements, and subjective, objective, assessment, and plan (SOAP) notes were retrieved from the big data center of TVGH. Images with an insufficiently high resolution were excluded to ensure data quality. 

Participants were enrolled only if they had (1) a diagnosis of DME (naïve, previously treated, or refractory); (2) macular edema, defined by a CST of >250 μm; (3) SRF or IRC, as indicated by spectral domain OCT; (4) BCVA between 0.05 and 1.5 (decimal) measured using a Snellen chart; and (5) an age of >20 years. Prospective participants were excluded if they had a history of any of the following ophthalmological diseases: (1) choroidal neovascularization due to AMD, RVO, uveitis, or any other inflammatory disease; (2) cataracts or clouded lenses; (3) glaucoma or any other neuropathy; (4) epiretinal membrane, vitreomacular traction disease, or any other maculopathy; or (5) corneal disease or degeneration. 

The decimal BCVA data of patients who received OCT imaging were retrieved from the SOAP notes on the closest date. These data were converted to the logarithm of the minimum angle of resolution (logMAR-BCVA), where a higher logMAR-BCVA indicates poorer VA.

### 2.2. Labeling of Retinal Features

After the images were collected and subjected to initial quality control, the BM data set was labeled by three experts, and an experienced ophthalmologist performed a final check to ensure that the label was correct. Using the image segmentation function of MATLAB (MathWorks, R2018a, Natick, MA, USA), the experts labeled each pixel as one of the following five retinal features, which served as biomarkers in this study: the neurosensory retina, EZ, retinal pigment epithelium (RPE), IRC, or SRF (Figure 2). Specifically, the neurosensory retina was defined as the area between the internal limiting membrane (ILM) and RPE. The EZ was defined as a hyperreflective linear structure between the external limiting membrane and the RPE. IRC was defined as any hyporeflective cyst between the ILM and EZ. Finally, SRF was defined as a hyporeflective cyst between the EZ and RPE.

### 2.3. Network Architectures

The structure of our fully convolutional segmentation network is similar to that of U-net [14], but we modified it to fit the task based on our expertise in medicine. Our model comprises an encoder and decoder, and some shortcuts between the encoder and decoder allow more information to be transmitted. IRC and SRF have various shapes, sizes and pixel intensities, and their boundaries are blurred. The encoder part of the conventional U-net was too shallow to extract enough features. To tackle these problems, we used the segmentation models proposed by Yakubovskiy, for which EfficientNet-B5 is the encoder backbone [29]. EfficientNet-B5 was proposed by Tan et al. It outperformed other deep learning architectures in image classification, which indicated that it could extract the most representative features of the image [30]. Here, EfficientNet-B5 is composed of seven distinct blocks used to extract features of different sizes. The detail of each block is presented in Appendix A
Table 1 shows the difference between U-net and the proposed model.

Our decoder is identical to that used by U-net. It comprises four decoder blocks, and each block is composed of one upsampling layer and two convolutional layers. The features extracted by the encoder blocks (block 1, 2, 3, 5) were concatenated to the decoder blocks to propagate high-resolution features. In the final part of the model, the sigmoid function is used to produce the segmentation result. The trainable parameters in the proposed model total 37 million. The architecture of the model is illustrated in Figure 3.

### 2.4. Experimental Setup

The OCT images in the BM data set were grouped into training (127 scans from 50 patients), validation (38 scans from 16 patients), and testing (38 scans from 17 patients) data sets. No single patient had images in two or more of these data sets. 

The segmentation network took OCT scans as inputs and output a probability map for each class. All the images were cropped to ensure uniform size. The images were augmented through slicing and flipping. Every image in the training data set was vertically sliced into eight slices and then flipped horizontally; this augmentation method is similar to that used by Roy et al. [19].

The segmentation network was trained, validated, and tested using a desktop computer with a central processing unit, an 8-core CPU (Core-i7-9700, Intel, Santa Clara, CA, USA), 16 GB memory, and a graphics processing unit (GPU) (RTX 2080 Ti, Nvidia, Santa Clara, CA, USA) with 12 GB graphics memory and 4352 CUDA cores. The operating system was Microsoft Windows 10. Table 2 presents the hyperparameters used for training.

The loss function was the averaged Dice loss, which was calculated as follows:(1)Loss of averaged Dice coefficient =1- averaged Dice coefficient
(2)$$\text{Averaged Dice coefficient }=\frac{\sum_{i=1}^{n}{\text{Dice coefficient }}_{i}}{n}$$
(3)$$\text{Dice coefficient }=\frac{2\sum_{i}^{N}p_{i}q_{i}}{\sum_{i}^{N}p_{i}^{2}+\sum_{i}^{N}q_{i}^{2}}$$
(4)Loss of Dice coefficient loss=1- Dice coefficient

Since the output of the segmentation network is a binary image, we can calculate the Sørensen–Dice coefficient using Equation (3). However, different numbers of pixels in different classes would affect the training results. If we do not compute the Dice coefficient for each class separately, but use Equation (4) to calculate the loss of Dice coefficient of the whole image, the model would tend to focus on classes with large numbers of pixels and ignore classes with fewer pixels. To alleviate this problem, the adopted averaged Dice coefficient (Equation (2)) is the average of the Dice coefficients across different classes, with equal contributions from each class regardless of the pixel number in each class. Therefore, the use of the loss of averaged Dice coefficient allowed us to deal with the class imbalance problem. 

### 2.5. Ablation Study

The ablation study was conducted to compare the performance between the conventional U-net and the proposed modified U-net based on the Dice coefficient. The keras-unet package, proposed by Karol Zak on github (https://github.com/karolzak/keras-unet, accessed on 21 May, 2022), was used as the conventional U-net. The hyperparameters of the conventional U-net and proposed U-net are shown in Table 2.

### 2.6. Quantification of Biomarkers

After the AI segmentation model was established, we applied the ECR data set to evaluate the associations of retinal features with BCVA. The ECR data set contained 3084 B-scan OCT images from 216 patients who had undergone intraocular lens procedures to prevent the effects of cataracts. Three retinal features, namely EZ disruption, IRC, and SRF, which were quantified by the AI segmentation model, and CST data that were collected from the OCT machine, were used to construct a linear regression model. IRC and SRF were quantified as the number of pixels, and the EZ was considered disrupted if the thickness of any part of the EZ was equal to 0.

### 2.7. Statistical Evaluation

To evaluate the performance of the segmentation network, we used the Sørensen–Dice coefficient between automatic and manual segmentation [31]. The Sørensen–Dice coefficient is equivalent to the F1 score, which measures both sensitivity and specificity and provides a holistic representation of algorithm performance. If no IRC or SRF was present in an image, the Sørensen–Dice coefficients for those biomarkers were not calculated for the image.

The ROC curve was used to evaluate performance in detecting EZ disruption, IRC, and SRF. The ROC curve for EZ disruption was calculated by changing the minimum thickness threshold for the EZ. The ROC curves for IRC and SRF were obtained by varying the threshold of the sum of the number of pixels attributed to IRC and SRF. The AUC is a parameter ranging from 0.5 to 1 and used to indicate an algorithm’s performance. AUC values of 0.5 and 1 indicate purely random and 100% accurate classification, respectively. 

Univariable linear regression, multivariable regression models, and correlational analyses were used to determine the association of the retinal features with the logMAR-BCVA. EZ disruption, IRC and SRF volume, and CST were chosen as predictors. All predictors were normalized prior to the regression. All analyses were conducted in MATLAB, and statistical significance was indicated if *p* < 0.05.

## 3. Results

### 3.1. Segmentation of Retinal OCT Images

To evaluate the performance of our automatic segmentation network, we compared automatic and manual segmentation results on the testing data set by using Sørensen–Dice coefficients. The manual segmentation was performed by two retinal specialists, and the interobserver agreement for retinal fluid was approximately 0.75. The Sørensen–Dice coefficients of the retinal structures and features are shown in Table 3. The Sørensen–Dice coefficients for the neurosensory retina and IRC were the highest and lowest, respectively. In general, all the Sørensen–Dice coefficients were larger than 0.75, and the Dice coefficient of the neurosensory retina was close to the maximum of 1.0 at >0.98. The average Dice coefficient was 0.86, with a standard deviation of 0.09. Compared with the conventional U-net, our proposed model achieved a higher Dice coefficient for EZ, IRC, and average Dice coefficient, and a mildly lower Dice coefficient for RPE and SRF.

Three randomly selected examples of raw images, manually segmented images (ground truth), and automatically segmented images (segmentation images) from the testing data set are displayed in Figure 4. The five retinal structures and features (neurosensory retina, EZ, RPE, IRC, and SRF) are color-coded and presented in the figure (middle and right columns for manual and automatic segmentation results, respectively). The red boxes indicate the differences in the images between the ground truth and automatic segmentation for IRC. Such differences may be caused by blurred boundaries (as displayed within the red boxes in the OCT slices in the left column). In addition to the IRC results, the results for the neurosensory retina, EZ, RPE, and SRF are highly consistent between the ground truth and segmentation images. 

### 3.2. Detection of Fluid and EZ Disruption

The ROC curve was used to evaluate the performance of the algorithm in detecting EZ disruption, IRC, and SRF. We also calculated the correlations of these features with logMAR-BCVA. Figure 5 shows the ROC curves for EZ disruption, IRC, and SRF in the testing data set. The performance of the algorithm was evaluated using the AUC. The AUC detecting EZ disruption by using the minimal thickness extracted from the automatic segmentation network was 0.88, and those for the detection of IRC and SRF from the sum of the fluid pixels were 0.96 and 1.00, respectively.

### 3.3. Associations with logMAR-BCVA

The associations between the chosen retinal features and BCVA were assessed using linear regression and correlational analyses. An OCT image was defined as indicating EZ disruption if the minimum thickness of the EZ was 0. Pixels with a probability >0.9 were summed to compute the volumes of IRC and SRF. Before linear regression, all the features were normalized through subtraction of the mean from each data point value and division of this difference by the standard deviation.

The correlations of the retinal features with BCVA are shown in Figure 6. The correlations for EZ disruption and IRC and CST volume were significant, with the strongest for EZ disruption. The volume of SRF was the most weakly correlated with BCVA, and this was the only correlation that was nonsignificant. Moreover, IRC was more correlated with BCVA than CST. However, the correlations between all these retinal features and BCVA were less than 0.5.

Table 4 presents the fitted coefficients from the linear regression model. In the univariable regression, EZ disruption, IRC volume, and CST were positive and significant predictors of logMAR-BCVA. EZ disruption had the largest correlation with logMAR-BCVA (slope = 0.428). Nevertheless, SRF volume had a nonsignificant correlation that was also the smallest. In the multivariable linear regression, all four features were significantly correlated with logMAR-BCVA. IRC and EZ disruption were positive predictors. EZ disruption had the largest effect on BCVA (slope = 0.402), as was the case in the univariable regression. SRF was a negative predictor, but its correlation with BCVA was the weakest (slope = −0.064). The multivariable findings for IRC and CST differed from their univariable counterparts: IRC was more strongly correlated with logMAR-BCVA than CST was in the univariable model, whereas this result was the opposite in the multivariable model.

## 4. Discussion

AI is a promising tool for processing imaging findings and clinical data. The application of AI to the treatment and diagnosis of DME, a leading cause of vision loss, helps clinicians to better assess the severity of any given case of DME. In this study, we used a small data set to train a segmentation model, which was then applied to a large data set. The proposed model could automatically detect and quantify EZ disruption, IRC, and SRF, with accuracies of AUC = 0.88, 0.96, and 1.0, respectively, and Sørensen–Dice coefficients of 0.81, 0.80, and 0.89, respectively. The correlations between the features extracted by the model and BCVA were determined; segmented EZ disruption and IRC were significantly correlated with logMAR-BCVA. These results demonstrate the practicability of deep learning neural networks for segmenting key DME-associated retinal features from OCT images for determining DME severity and treatment progress.

Our method performed better than others in the literature. For example, Lu et al. achieved a mean Sørensen–Dice coefficient of 0.73 and mean AUC of 0.99 in the detection of retinal fluid when they applied their method to the RETOUCH database [32]. Lee et al. obtained a mean Sørensen–Dice coefficient of 0.729 when using their method to detect IRC [33]. Our method performed better at segmenting and detecting retinal fluid, with a Sørensen–Dice coefficient of 0.80. Orlando et al. applied their method to detect EZ disruption and attained a Sørensen–Dice coefficient of 0.89 among patients with AMD but an AUC of only 0.67 among patients with only DME or RVO.

Previous studies have reported that EZ integrity is the most strongly correlated feature with VA in patients with DME [4,5]. Our segmentation model can accurately detect EZ disruption, SRF, and IRC, and the extent of EZ disruption identified by our model was most strongly correlated with logMAR-BCVA.

In the current state of the art, physicians and researchers use CST to assess DME severity and track treatment progress. Nevertheless, we found that EZ disruption and IRC were more strongly correlated with BCVA than was CST. Increases in CST are primarily caused by IRC and SRF. This may explain why CST was more strongly correlated with BCVA than IRC was in the multivariable regression.

Some studies have revealed that the presence of SRF did not necessarily worsen BCVA and that tolerance toward SRF may preserve a patient’s BCVA, unlike total dryness of macula [34,35]. Our study found SRF to be a negative predictor of logMAR-BCVA, implying that some residual SRF may help to preserve BCVA, a finding consistent with those of SRF tolerance in prior studies.

Our study has some limitations. First, our data sets were limited in scope; this limited the performance of our model, which nonetheless had similar accuracy as its counterparts in the literature [17,20,33,36]. Second, the OCT images in our study featured only two-dimensional slices from the central macula, meaning that IRC and SRF were not detected if they were outside the central area. In future studies, we will apply this model to three-dimensional volumetric OCT images for a more comprehensive analysis. Third, the data set covered only patients with DME, and our model may thus be inapplicable to other retinal diseases. In future studies, will apply our model to data for other retinal diseases.

## 5. Conclusions

We formulated and validated a fully automatic segmentation model that can effectively quantify the OCT biomarkers of EZ disruption, IRC, and SRF in patients with DME. These OCT biomarkers are more correlated with VA than is CST, which is presently analyzed by clinicians and researchers from retinal OCT images. Our novel deep learning model aids in the clinical determination of DME severity and treatment progress. Future applications to three-dimensional OCT images are warranted to improve the model.

## Figures and Tables

**Figure 1 biomedicines-10-01269-f001:**
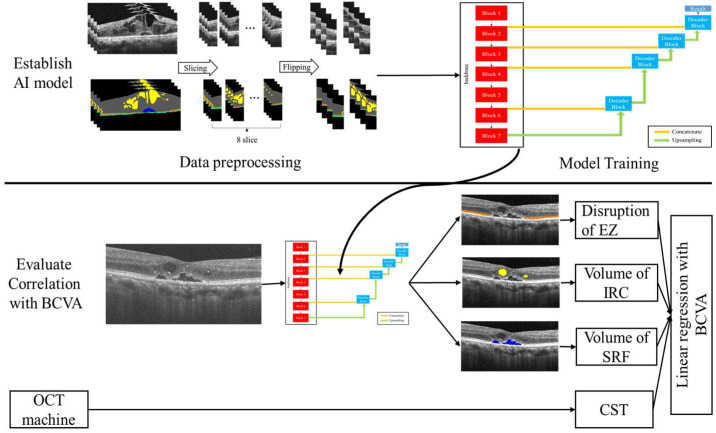
The overview of the experiment. AI: artificial intelligence; BCVA: best corrected visual acuity; EZ: ellipsoid zone; IRC: intraretinal cystoid fluid; SRF: subretinal fluid; CST: central subfield thickness.

**Figure 2 biomedicines-10-01269-f002:**
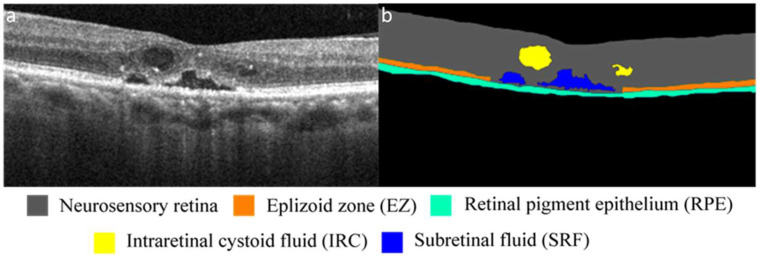
(**a**) OCT image of retina of patient with DME. (**b**) Segmentation of OCT image indicating, through color coding, different retinal structures and features.

**Figure 3 biomedicines-10-01269-f003:**
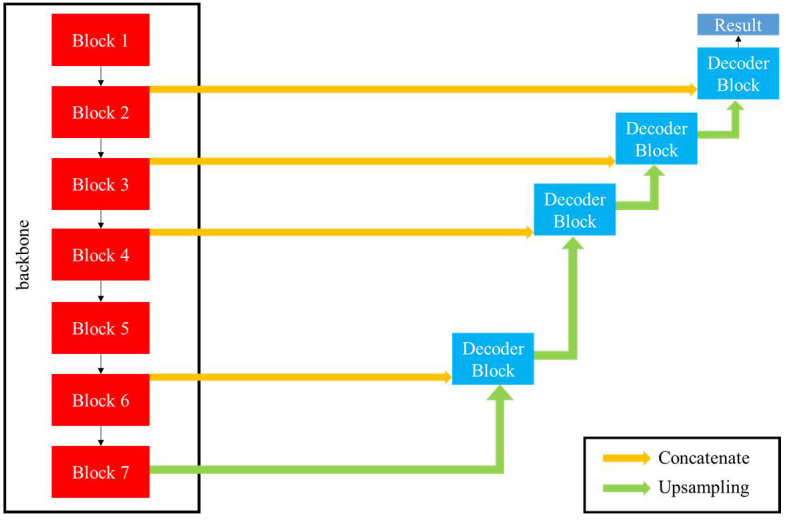
Architecture of proposed model. Decoder block is the same as that of U-net, and the backbone is EfficientNet-B5, which is composed of seven blocks.

**Figure 4 biomedicines-10-01269-f004:**
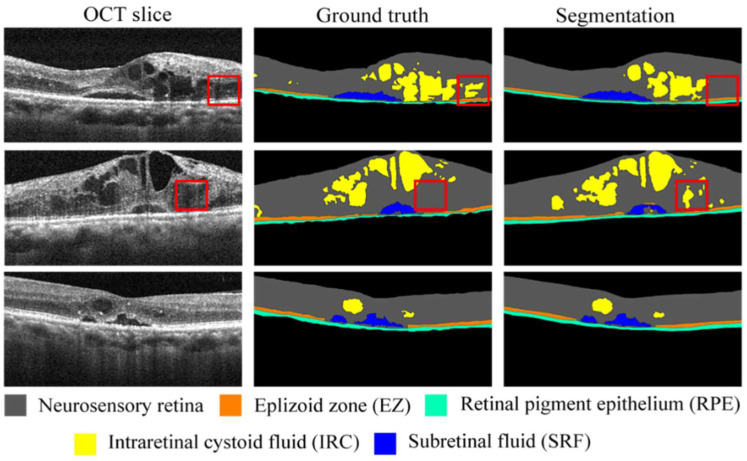
Randomly selected OCT images of three patients with DME (left column), their corresponding ground truth images (middle column), and segmented images (right column). Red boxes in first and second rows indicate differences between ground truth and segmentation.

**Figure 5 biomedicines-10-01269-f005:**
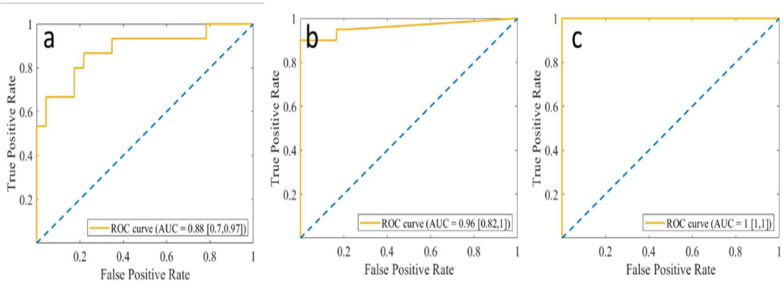
ROC curves of (**a**) EZ disruption, (**b**) IRC, and (**c**) SRF. Blue dotted lines indicate AUC = 0.5. EZ: ellipsoid zone; IRC: intraretinal cystoid fluid; SRF: subretinal fluid.

**Figure 6 biomedicines-10-01269-f006:**
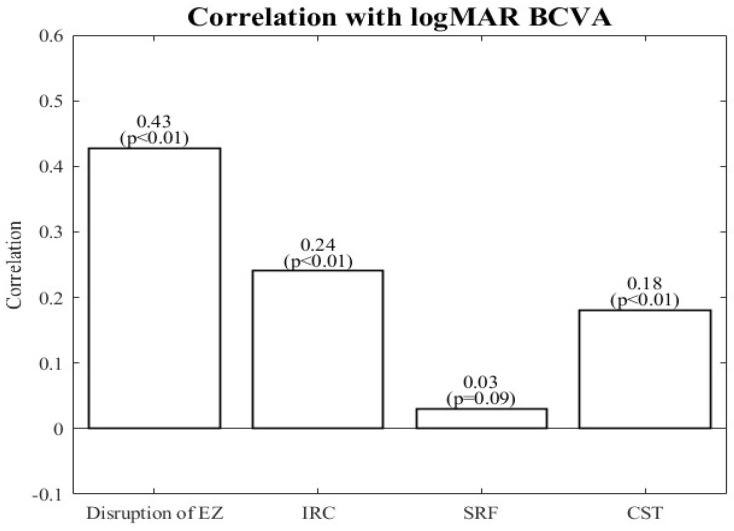
Correlation of retinal features with BCVA. BCVA: best corrected visual acuity; EZ: ellipsoid zone; IRC: intraretinal cystoid fluid; SRF: subretinal fluid; CST: central subfield thickness.

**Table 1 biomedicines-10-01269-t001:** The difference between U-net and proposed model.

	U-Net [14]	Proposed
Encoder	8 convolution layers with 4 max pooling layers	EfficientNet-B5 [30] (7 blocks) ^1^
Decoder	8 convolution layers with 4 up-convolution layers	8 convolution layers with 4 up-convolution layers

^1^ The detail of each block is presented in Appendix A.

**Table 2 biomedicines-10-01269-t002:** Hyperparameters for training.

Hyperparameter	Selected Value
Backbone of encoder	EfficientNet-B5
Loss function	Loss of averaged Dice coefficient
Optimizer	Adam [31]
Learning rate	1 × 10^−4^
Batch size	10
Epoch	50

**Table 3 biomedicines-10-01269-t003:** Dice coefficient of each retinal structure and feature segmented by U-net and proposed model. The higher Dice coefficient is marked in bold.

Retinal Features	Dice Coefficient	
	U-Net	Proposed
Neurosensory retina	0.98 ± 0.02	0.98 ± 0.01
EZ ^1^	0.80 ± 0.09	**0.81 ± 0.08**
RPE ^2^	**0.83 ± 0.04**	0.82 ± 0.04
IRC ^3^	0.61 ± 0.22	0.80 ± 0.08
SRF ^4^	**0.9 ± 0.02**	0.89 ± 0.04
Average	0.84 ± 0.15	**0.86 ± 0.09**

^1^ EZ: ellipsoid zone; ^2^ RPE: retinal pigmented epithelium; ^3^ IRC: intraretinal cystoid fluid; ^4^ SRF: subretinal fluid.

**Table 4 biomedicines-10-01269-t004:** Linear regression results for VA (logMAR-BCVA) in relation to OCT-derived retinal features (EZ disruption, IRC, and SRF).

Features	Univariable			Multivariable		
	β	SE	*p*-Value	β	SE	*p*-Value
**Disruption of EZ ^1^**	0.428	0.016	<0.001	0.413	0.021	<0.001
**IRC ^2^**	0.240	0.017	<0.001	0.083	0.022	<0.001
**SRF ^3^**	0.031	0.018	0.088	−0.064	0.016	<0.001
**CST ^4^**	0.181	0.018	<0.001	0.110	0.022	<0.001

^1^ EZ: ellipsoid zone; ^2^ IRC: intraretinal cystoid fluid; ^3^ SRF: subretinal fluid; ^4^ CST: central subfield thickness.

## Data Availability

The data presented in this study are available upon request from the corresponding author.

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
