# Peer review of "Automatic Segmentation of Retinal Fluid and Photoreceptor Layer from Optical Coherence Tomography Images of Diabetic Macular Edema Patients Using Deep Learning and Associations with Visual Acuity"

_biomedicines, 2022, doi:10.3390/biomedicines10061269_

Round 1
Reviewer 1 Report
Dear Author(s),
Thanks for submitting your manuscript on Biomedicines.
The AI will surely change many medicine sectors and one of the more involved will be the ophthalmology field. In particular, the great amount of imaging data obtained every year will be screened and rapidly elaborated by the artificial intelligence. In this sense, the AI application in diabetic retinopathy is absolutely a path to follow. In your article, I do not find any new insights on the OCT parameters used to assess the severity of the disease (ellissoid disruption, SFR, IRF, RPE alterations). On the other hand, I have read with great interest the high rate of detection, measured through the mean Sørensen–Dice coefficient. Furthermore, I would underline the great clearness of the entire text, which let the reader easily understand the all the process used. Probably for a formatting problem, I see the figure 5 partially hiding a part of the text. After this minor issue, I would recommend this paper for publication.
Reviewer 2 Report
The authors presented a deep learning-based technique for segmentation in DME images. I have the following concerns:
1) Abstract is missing with how the existing methods are lacking, and how the proposed method will help in such diagnosis.
2) Introduction is missing an important literature review (Latest 2020~2022), it is not good to conclude a research article with just 25 references. The introduction is really weak (rewrite required).
3) The authors fail to describe the proposed method, just one diagram Figure 3 is not enough to explain pixel-wise semantic segmentation, there is no explanation of the connectivity, layers, feature map sizes, what is inside each encoder-decoder blocks?
4) how the model is different from U-Net conventional? provide a table describing the architectural differences. Explain why to modify an existing architecture, is there any specific reason ? That hypothesis and theoretical discussion is missing.
5) An ablation study with U-Net is required (experiment+ description)
6) How do the authors handle the class imbalance ? provide proper information in the paper, also provide the explanation of the loss function used in experiments with reason.
7) Describe the hardware that is used for training and testing fo the models
8) how many numbers of trainable parameters are consumed by your network?
Overall comment: the proposed method is not properly described, you can follow this paper on how to describe the proposed method theoretically and mathematically :
Detecting retinal vasculature as a key biomarker for deep Learning-based intelligent screening and analysis of diabetic and hypertensive retinopathy
Round 2
Reviewer 2 Report
The authors correctly responded to my comments. I vote for acceptance of this paper in current form